# Divergent roles of macrophage subsets, FoxP3, and IL-17A in HSV-1–induced CNS pathology

Ujjaldeep Jaggi, Satoshi Hirose, Shaohui Wang, Homayon Ghiasi [ID]*

Center for Neurobiology and Vaccine Development, Ophthalmology Research, Department of Surgery, CSMC, Los Angeles, California, United States of America

* ghiasih@CSHS.org

## Abstract

As a central player in neuroinflammation, macrophages play multifaceted roles such as antigen presentation, phagocytosis, production of cytokines/chemokines, and growth/neurotrophic factors. Our previous work demonstrated that ocular infection with a recombinant herpes simplex virus type 1 (HSV-1) expressing interleukin-2 (HSV-IL-2) causes CNS pathology, independently of macrophages in different mouse strains. In contrast, wild type (WT) HSV-1 infection induces CNS demyelination in a macrophage-dependent manner. Therefore, in this study, we have two mouse models infected with either HSV-IL-2 or WT HSV-1 to examine the outcome of the absence of IL-17A, FoxP3, macrophages, or combined macrophage and FoxP3 depletion on CNS demyelination. Our data reveals several notable findings: deletion of FoxP3 alone in mice infected with either HSV-IL-2 or WT HSV-1 did not induce CNS demyelination. However, combined depletion of macrophages and FoxP3 in HSV-IL-2-infected mice triggered CNS demyelination, whereas the same combined depletion in WT HSV-1 infection prevented demyelination. Additionally, macrophage depletion alone in WT HSV-1-infected mice induced CNS demyelination, highlighting the non-redundant protective role of macrophages in this model. To further elucidate the role of macrophages in CNS demyelination, we investigated which macrophage subtype is responsible for modulating demyelination using M1 and M2 knockout mice. Our results indicate that M1 macrophages are key drivers of plaque formation, as infection with either HSV-IL-2 or WT HSV-1 failed to cause CNS demyelination in the absence of M1 macrophages. Conversely, M2-deficient mice exhibited demyelination, suggesting a protective role for M2 macrophages. Finally, depletion of macrophages in IL-17A-deficient mice infected with HSV-IL-2 did not restore CNS demyelination, indicating that, unlike the macrophage-FoxP3 double depletion in the HSV-IL-2 model, the IL-17A–macrophage absence is beneficial. Taken together, these findings highlight the distinct and non-redundant roles of FoxP3, IL-17A, and macrophage subsets in modulating CNS pathology during HSV-1 infection and suggest that targeting M1 macrophage activation may be a promising strategy for limiting demyelination.

**Data availability statement:** All relevant data are in the manuscript and its supporting information files.

**Funding:** This work was supported by Public Health Service grants RO1EY024649, RO1EY029677, and RO1EY013615 from the National Eye Institute to HG. The funders had no role in study design, data collection and analysis, decision to publish, or preparation of the manuscript.

**Competing interests:** The authors have declared that no competing interests exist.

## Author summary

Demyelinating diseases include a range of immunopathologic diseases in which myelin, the covering of nerve cell fibers in the central nervous system (CNS), is destroyed. While the roots of demyelination are not well understood, one hypothesis is that it may result from autoimmunity to CNS antigens triggered by environmental factors, with the activated immune response leading to myelin destruction. Multiple sclerosis (MS) is one of the major diseases associated with myelin sheath degradation. In the optic nerve, demyelination and inflammation cause visual and neurologic dysfunction associated with optic neuritis (ON), an initial manifestation and early prognostic indicator of MS in young adults. Macrophage plays a vital role in maintaining immune homeostasis in the CNS by preventing the development of autoaggressive T cells, which are responsible for both autoimmunity and tolerance. In our current study, we investigated the role(s) of $T_H17$ and Treg in the presence and absence of macrophages on CNS demyelination. We demonstrated the essential role of M2 macrophages compared with M1 macrophages in maintaining immune homeostasis and effectively preventing CNS demyelination in ocularly infected mice.

## Introduction

Macrophages perform a wide range of functions in the immune system, with their polarization state dictated mainly by the surrounding microenvironment. They can adopt a classically activated M1 (pro-inflammatory) or alternatively activated M2 (anti-inflammatory) phenotype. In the context of multiple sclerosis (MS), the activated M1 macrophages are associated with enhanced antigen presentation and the production of pro-inflammatory cytokines and cytotoxic molecules [1,2]. In contrast, activated M2 macrophages have been linked to remyelination, primarily by promoting oligodendrocyte progenitor cell (OPC) differentiation through the secretion of growth factors [1]. Following ocular infection of naive mice with HSV-1, macrophages represent the predominant infiltrating immune cell into the eye [3–5], and may play a dual role in both exacerbation and control of acute and chronic inflammation [6–9]. Previously, we have shown that depletion of macrophages, but not other immune cell types, induces demyelination in the brain, optic nerves, and spinal cord of ocularly infected mice. Notably, this demyelination is independent of both the virus strain and the mouse genetic background [10].

Macrophage-derived IL-12p70, composed of IL-12p35 and IL-12p40, is crucial for protecting against HSV-1–induced CNS demyelination, as its absence or macrophage depletion leads to pathology reversible by IL-12 restoration [11]. In contrast, HSV-1 expressing IL-2 induces demyelination even in immunocompetent mice [11,12], underscoring a unique pathogenic role for IL-2 in driving demyelination. IL-2 plays a leading role in regulating the adaptive immune response and signals through its heterotrimeric receptor consisting of α (IL-2rα, CD25), β (IL-2rβ, CD122) and γ

PLOS Pathogens

(IL-2rγ, CD132) chains [13,14]. We previously looked at the role of these receptors in HSV-IL-2-induced CNS demyelination using IL-2rα$^{-/-}$, IL-2rβ$^{-/-}$, and IL-2rγ$^{-/-}$ mice. Our findings revealed demyelination in the brain, spinal cord, and optic nerves of IL-2rα$^{-/-}$ and IL-2rβ$^{-/-}$ mice but not in the CNS of IL-2rγ$^{-/-}$ infected mice [15]. In line with the absence of CNS demyelination in IL-2rγ$^{-/-}$, previously it was reported that IL-2rγ$^{-/-}$ mice lack group 2 innate lymphoid cells (ILC2s), and IL-2 is known to be a critical regulator of ILC2 development and function [16]. Infection of ILC1$^{-/-}$, ILC2$^{-/-}$ and ILC3$^{-/-}$ mice with HSV-IL-2 has shown the absence of demyelination in ILC2$^{-/-}$ mice but not ILC1$^{-/-}$ or ILC3$^{-/-}$ mice [15]. These results indicate that, ILC2s play a negative role in the outcomes of CNS pathology in HSV-IL-2 infected mice and may be the key mediators of IL-2-induced demyelination. ILC2s are known to interact with macrophages [17], thus, to examine differences in CNS demyelination between macrophage-depleted mice following infection with WT HSV-1 versus mice infected with HSV-IL-2 virus, we specifically examined the roles of M1 and M2 macrophages in infected mice. Our results demonstrate that in both our models of CNS demyelination, M2 macrophages play an indispensable role in orchestrating self-tolerance and prevention of autoimmunity.

Previously, we have demonstrated that T cells contribute to HSV-IL-2-induced CNS demyelination [10,18,19], and both the forkhead/winged helix transcription factor gene P3 (FoxP3) and T$_H$17 are critical immune regulators in the contexts of autoimmunity, allergy, and infectious disease [20–31]. In our current report, we investigated the potential crosstalk between FoxP3 or T$_H$17 cells with macrophages in CNS demyelination. Our findings indicate that, in the HSV-IL-2 model of CNS pathology, demyelination occurring in the absence of M1 macrophages and proceeds independent of FoxP3$^+$ cells. In contrast, in WT HSV-1 infected mice, demyelination appears to be dependent on FoxP3$^+$ cells. Furthermore, the absence of T$_H$17 cells did not influence CNS demyelination in a macrophage-dependent manner.

## Results

### Inter-relationship of FoxP3 and macrophages in CNS demyelination in infected mice

Previously, we reported that both CD4$^+$ and CD8$^+$ T cells contribute to HSV-IL-2-induced CNS demyelination [18]. We also demonstrated that macrophage depletion induces CNS demyelination in WT mice infected with WT HSV-1, and FoxP3 depletion blocked demyelination in macrophage-depleted mice [10]. To further assess the impact of depleting FoxP3, macrophages, or both on protection from CNS demyelination in HSV-IL-2 infected mice, FoxP3$^{DTR}$ mice were infected with HSV-IL-2 or WT HSV-1 and subjected to depletion of FoxP3, macrophages, or both, as detailed in Materials and Methods. Mock depleted FoxP3$^{DTR}$ mice were infected similarly and served as controls. Demyelination in optic nerves, spinal cord, and brain was examined on day 14 PI using Luxol Fast Blue (LFB) staining. Representative photomicrographs of optic nerve, spinal cord, and brain sections from HSV-IL-2 or WT HSV-1 infected mice are shown in Fig 1. Consistent with our previous reports [11,18], conspicuous demyelination was observed in the optic nerve, spinal cord, and brain of HSV-IL-2–infected, FoxP3$^{DTR}$ mice (Fig 1, mock depleted FoxP3$^{DTR}$ mice, HSV-IL-2 infected column, arrows). However, similar to our previous studies [32], no demyelination was detected in optic nerve, spinal cord, and brain sections from WT HSV-1 infected mice (Fig 1, mock depleted, WT HSV-1 infected column). Our results have shown that no demyelination plaques were detected in the optic nerve, spinal cord, or brain of HSV-IL-2 infected or WT HSV-1 infected mice after FoxP3 depletion, indicating that depletion of FoxP3 alone does not affect demyelination (Fig 1, FoxP3-depleted, HSV-IL-2 and WT HSV-1 infected columns).

When mice were depleted of both FoxP3 and macrophages and infected with HSV-IL-2, demyelination was observed in optic nerve, spinal cord, and brain of infected mice (Fig 1, Mφ-FoxP3-depleted, HSV-IL-2 infected column, see arrows). In contrast, in WT HSV-1 infected mice, combined depletion of FoxP3 and macrophages prevented demyelination in optic nerve, spinal cord, and brain of infected mice (Fig 1, Mφ-FoxP3-depleted, WT HSV-1 infected column). Furthermore, in our WT control mouse model, macrophage depletion alone, followed by infection with WT HSV-1 resulted in demyelination in optic nerve, spinal cord, and brain (Fig 1, Mφ-depleted, WT HSV-1 infected column, see arrows), consistent with our previous findings [10].

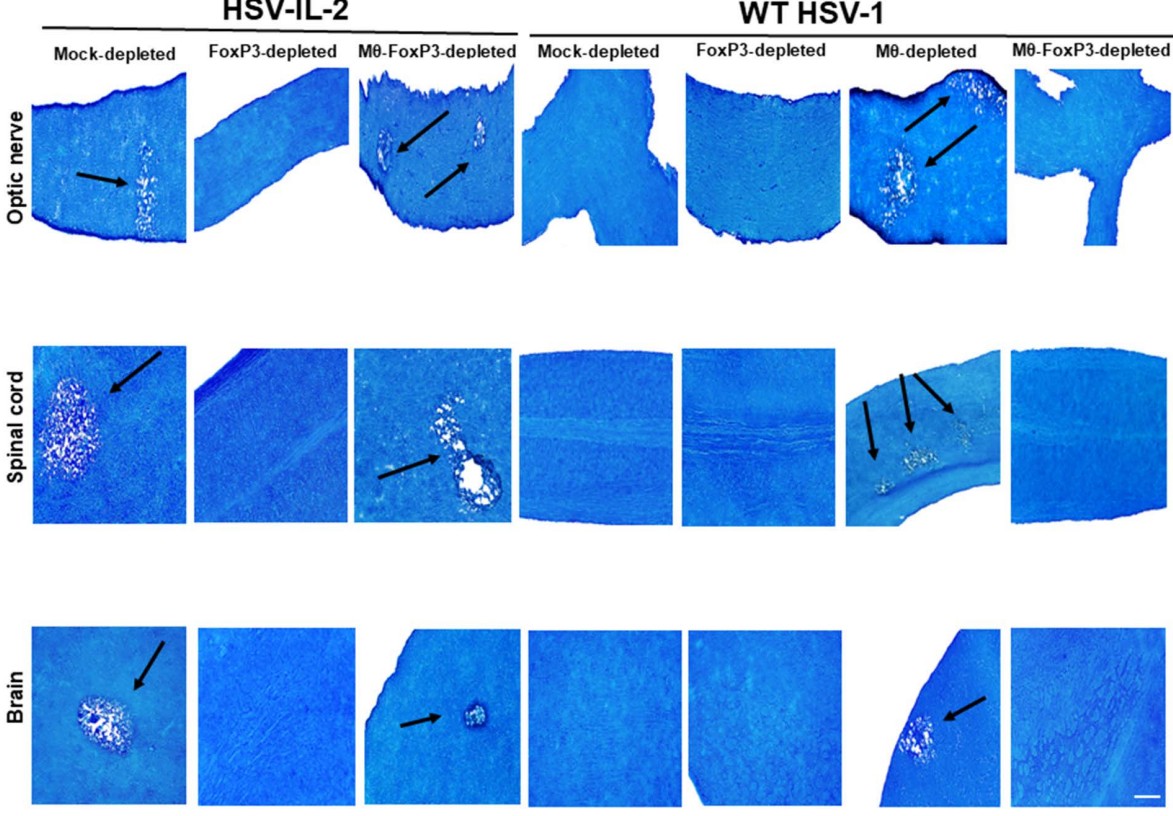

**Fig 1. Role of FoxP3 and macrophages in the induction of demyelination in CNS of depleted mice.** Five female FoxP3^DTR mice were depleted of both FoxP3 and macrophage, or mock-depleted, and then infected ocularly with 2 X 10^5 pfu/eye of either HSV-IL-2 or WT HSV-1 control virus (strain McKrae). Representative photomicrographs of optic nerve, spinal cord, and brain sections on day 14 PI from infected mice are shown. Arrows indicate areas of demyelination (i.e., plaques). Scale bar denotes 100 μm.

Thus, our results demonstrate that FoxP3 alone is sufficient to prevent CNS demyelination in HSV-IL-2–infected mice, while the combined depletion of FoxP3 and macrophages promotes demyelination, suggesting a synergistic pathogenic interaction between these two components. In contrast, during WT HSV-1 infection, macrophages appear to play a protective role against CNS demyelination in a FoxP3-dependent manner.

## Enumeration of regulatory T cells in infected mice

Both CD4+ and CD8+ T cells play key roles in affecting demyelination in animal models of MS [33–38]. To directly assess whether the frequency of both CD4+ and CD8+ regulatory T cells (Tregs) might be altered during HSV-IL-2 infected mice, compared with WT HSV-1 control mice, mice were infected ocularly with either HSV-IL-2 or WT HSV-1 as described above. Infected mice were sacrificed on day 3 PI, and splenocytes were harvested, stained, and analyzed by flow cytometry to assess for the presence of CD4 and CD8 Foxp3 cells, as described in Materials and Methods. We found an increase in the individual populations of CD4+FoxP3+ cells in the spleen of HSV-IL-2 infected mice compared to HSV-1 infected mice (Fig 2A, representative dot plots). These differences were statistically significant (Fig 2C, p = 0.04 Student's t Test). Increased CD8+FoxP3+ cells were also found in the spleen of HSV-IL-2 infected mice compared with the spleen of HSV-1 infected mice (Fig 2B, representative dot plots). Splenocytes from HSV-IL-2 infected mice had a significantly higher percentage of CD8+FoxP3+ cells compared to splenocytes from HSV-1 infected mice (Fig 2B, P < 0.01). Our data indicate that

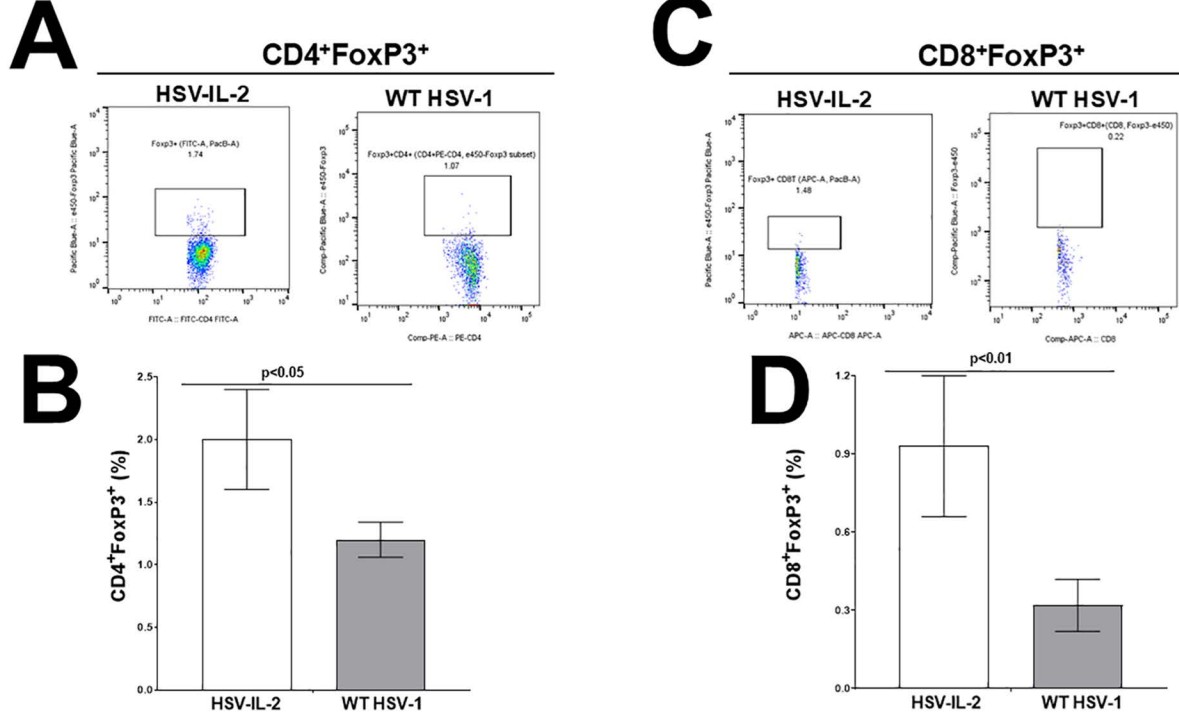

**Fig 2. Expansion of CD4‡FoxP3± and CD8‡FoxP3± T cells from splenocytes of HSV-IL-2-infected mice.** Spleens from WT C57BL/6 mice infected with HSV-IL-2 or WT HSV-1 strain McKrae were harvested 3 days PI as described in Materials and Methods. Spleens were dissociated into a single cell suspension and were stained with antibodies against CD3/CD4/CD8/FoxP3, and the percentage of positive cells was determined by flow cytometry. Panels: (**A**) Representative dot plots from CD3+CD4+FoxP3+ cells from HSV-IL-2 and WT HSV-1 infected cells; (**B**) Representative dot plots from CD3+CD8+FoxP3+ cells from HSV-IL-2 and WT HSV-1 infected mice; (**C**) Quantification of the mean number of CD3+CD4+FoxP3+ cells from HSV-IL-2 and WT HSV-1 infected cells; and (**D**) Quantification of the mean number of CD3+CD8+FoxP3+ cells from HSV-IL-2 and WT HSV-1 infected cells. Data are based on 3 mice spleens ± SEM.

the increased frequency of both CD4 and CD8 Tregs in the spleens of HSV-IL-2 infected mice is attributable to the presence of IL-2 of the HSV-IL-2 virus; hence, Tregs expand and may contribute to virus-induced CNS demyelination. Previously, it was shown that higher expression of FoxP3 in Treg cells is dependent on IL-2 [39,40]. Thus, we conclude that in the presence of IL-2-expressing HSV-1, T cells express higher FoxP3 levels than when mice are infected with WT HSV-1, and HSV-IL-2-induced CNS demyelination is dependent on FoxP3.

## Depletion of macrophages in the absence of IL-17A does not cause CNS demyelination in infected mice

The above results suggest that FoxP3 in HSV-IL-2 infected mice is contributing to CNS demyelination in a macrophage-dependent manner. In contrast, in WT HSV-1 infected mice, FoxP3 appears to protect against CNS demyelination in the absence of macrophages (see Fig 1 above). Similar to FoxP3-depletion, recently we reported that IL-17A-/- mice are refractory to HSV-IL-2-induced CNS demyelination [32]. To determine whether, similar to FoxP3-macrophage co-depletion, depletion of macrophages in IL-17A-/- mice could restore demyelination in HSV-IL-2 infected mice, female IL-17A-/- mice were depleted of macrophages as described in Materials and Methods. WT C57BL/6 control mice were similarly depleted of macrophages and infected with WT HSV-1 control. Demyelination in the optic nerve, spinal cord, and brain was assessed on day 14 PI using LFB staining, as described above.

Representative photomicrographs of optic nerve, spinal cord, and brain sections from HSV-IL-2 or WT HSV-1 infected IL-17A -/- (Mϕ depleted) and WT (Mϕ depleted) mice are shown in Fig 3. In HSV-IL-2 infected mice, no demyelination was

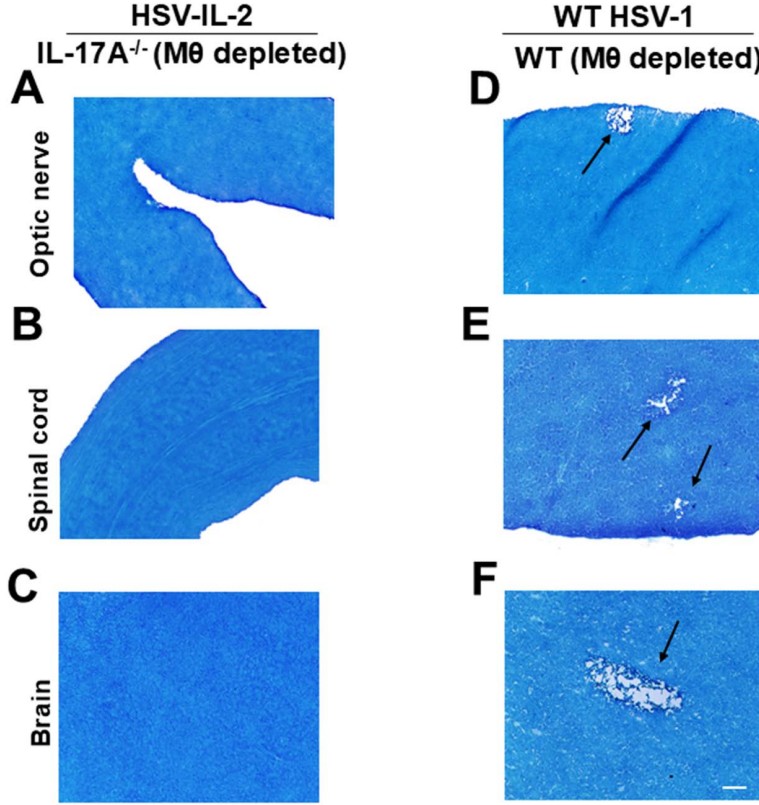

**Fig 3. Role of IL-17A and macrophages in the induction of demyelination in the CNS of depleted mice.** Female IL-17A$^{-/-}$ mice or WT C57BL/6 mice were macrophage-depleted and infected ocularly with HSV-IL-2 or WT HSV-1 control virus (strain McKrae). Tissues were harvested on day 14 PI. **(A-C)** represents photomicrographs of optic nerve, spinal cord, and brain sections infected with HSV-IL-2. **(D-F)** represents photomicrographs of optic nerve, spinal cord, and brain sections infected with WT HSV-1. Arrows indicate areas of demyelination (i.e., plaques). Scale bar denotes 100 μm.

detected in optic nerve (Fig 3A), spinal cord (Fig 3B), or brain (Fig 3C) of infected IL-17A $^{-/-}$ (Mϕ depleted) mice, but as expected and similar to Fig 1 above, demyelination plaques were present in optic nerve (Fig 3D), spinal cord (Fig 3E), and brain (Fig 3F) of WT HSV-1 infected WT (Mϕ depleted) mice. Similar to IL-17A$^{-/-}$ mice that did not develop CNS demyelination following HSV-IL-2 infection, macrophage-depleted IL-17A$^{-/-}$ mice infected with WT HSV-1 also showed no demyelination. These results suggest that, in the HSV-IL-2 infection model, combined depletion of FoxP3$^{+}$ regulatory T cells and macrophages promotes CNS demyelination, indicating that FoxP3-induced protection is macrophage-dependent. In contrast, depletion of macrophages in the absence of IL-17A does not induce demyelination, suggesting that the simultaneous absence of IL-17 and macrophages is protective in this model.

## Size and number of plaques in the CNS of mice infected with HSV-IL-2 and WT HSV-1 in the absence of macrophages

To quantify the extent of demyelination in the mice described above, we counted the number of demyelinated lesions in the optic nerves, spinal cord, and brain sections. Additionally, we measured the size of the observed demyelination plaques in these CNS regions. This analysis was performed on tissues from five female mice each of mock-depleted, FoxP3-depleted, Mϕ-depleted, and Mϕ-FoxP3-depleted mice infected with HSV-IL-2 and Mϕ-depleted, and Mϕ-FoxP3-depleted in WT HSV-1 control viruses, as we described previously [18,41]. The data are presented as the number of

sections showing demyelination plaques per total stained sections, along with the area of demyelination per section (Fig 4). Consistent with the observations in Fig 1, no plaques were detected in the optic nerves, brain, and spinal cord of HSV-IL-2-infected mice depleted of FoxP3, nor were they in mice infected with WT HSV-1 (Fig 4). Furthermore, as shown, combined depletion of Mφ-FoxP3 did not result in plaque formation, and there were no significant differences in the number/size of plaques across the optic nerves, brain, and spinal cord. (Fig 4A–4F, P > 0.05).

The number of plaques per optic nerve section of mice infected with the two viruses was similar (Fig 4A, P > 0.05). In contrast, brain sections from HSV-IL-2 infected mice depleted of macrophages exhibited a significantly higher number of plaques compared to other groups (Fig 4B, P < 0.05). Although macrophage-depleted mice infected with either virus showed a greater number of plaques than mock-depleted or combined macrophage and FoxP3-depleted groups, these differences did not reach statistical significance in spinal cord sections (Fig 4C, P > 0.05). Regarding plaque size, mock-depleted mice infected with either virus tended to have larger plaques in the optic nerve than other groups, but these differences were not statistically significant (Fig 4D, P > 0.05). The size of the plaques in the brain (Fig 4E) and spinal cord (Fig 4F) of all groups of infected mice with demyelination was similar to each other (Fig 4E and 4F, P > 0.05).

Overall, except for the lower number and size of plaques in the optic nerve but not in the spinal cord and brain, the absence of macrophages did not affect the number or size of plaques in the optic nerve, brain or spinal cord of infected mice compared with WT HSV-1. Thus, this study suggests that macrophages contribute to protection against CNS demyelination.

**M1 macrophages contribute to CNS demyelination**

Similar to T cells, macrophages have been divided to at least two groups M1 (pro-inflammatory) and M2 (anti-inflammatory), each exhibiting distinct functions both *in vitro* and *in vivo* [42–53]. Our depletion studies suggest that macrophages contribute to protection against CNS demyelination following ocular infection with WT HSV-1 [10]. To investigate whether M1 or M2 macrophage subsets are contributing to CNS demyelination and whether the pattern of demyelination differs between HSV-IL-2 and WT HSV-1 infected mice, we ocularly infected mice lacking M1 (i.e., M1$^{-/-}$) or M2 (i.e., M2$^{-/-}$) with either HSV-IL-2 or WT HSV-1 strain KOS as described in Materials and Methods and above. In this study, due to the known susceptibility of M1$^{-/-}$ mice to the virulent HSV-1 strain McKrae [52,53], we infected both groups of mice with the weakly virulent HSV-1 strain KOS. Similarly, HSV-IL-2 virus, unlike its parental strain McKrae, is also weakly virulent [54]. We monitored demyelination in the optic nerve, spinal cord, and brain of infected mice on day 14 PI. M1$^{-/-}$ mice infected with either HSV-IL-2 virus or WT HSV-1 showed no signs of demyelination in any CNS tissue examined (Fig 5). In contrast, M2$^{-/-}$ mice infected with either virus exhibited prominent demyelination across all CNS regions examined (Fig 5). These findings indicate that M1 macrophages are key effectors driving CNS demyelination in both HSV-IL-2 and WT HSV-1 infection models, while M2 macrophages likely play a protective role.

**Discussion**

Multiple sclerosis (MS) is a chronic inflammatory disorder of the central nervous system (CNS), characterized by immune-mediated demyelination and neurodegeneration. In MS, circulating monocytes infiltrate the CNS and differentiate into tissue-resident macrophages, which accumulate in large numbers within demyelinated lesions. These macrophages play a significant role in driving disease progression by promoting demyelination and tissue destruction [55]. One of the most remarkable characteristics of macrophages is their plasticity and heterogeneity, and their division into M1/M2 macrophages [42,43,56]. Previously we reported that administration of WT mice with M-CSF or IL-4, or their infection with a recombinant HSV-1 expressing IL-4 (HSV-IL-4), pushes macrophages toward an M2 response, whereas injection of mice with IFN-γ or their infection with an IFN-γ expressing recombinant virus (HSV-IFN-γ) enhances an M1 response [57,58]. Using M2-knockout mice and M2 transgenic mice (overexpressing M2 macrophages), we observed that elevated M2 macrophage levels correlate with enhanced phagocytosis, increased primary virus replication, and increased latency.

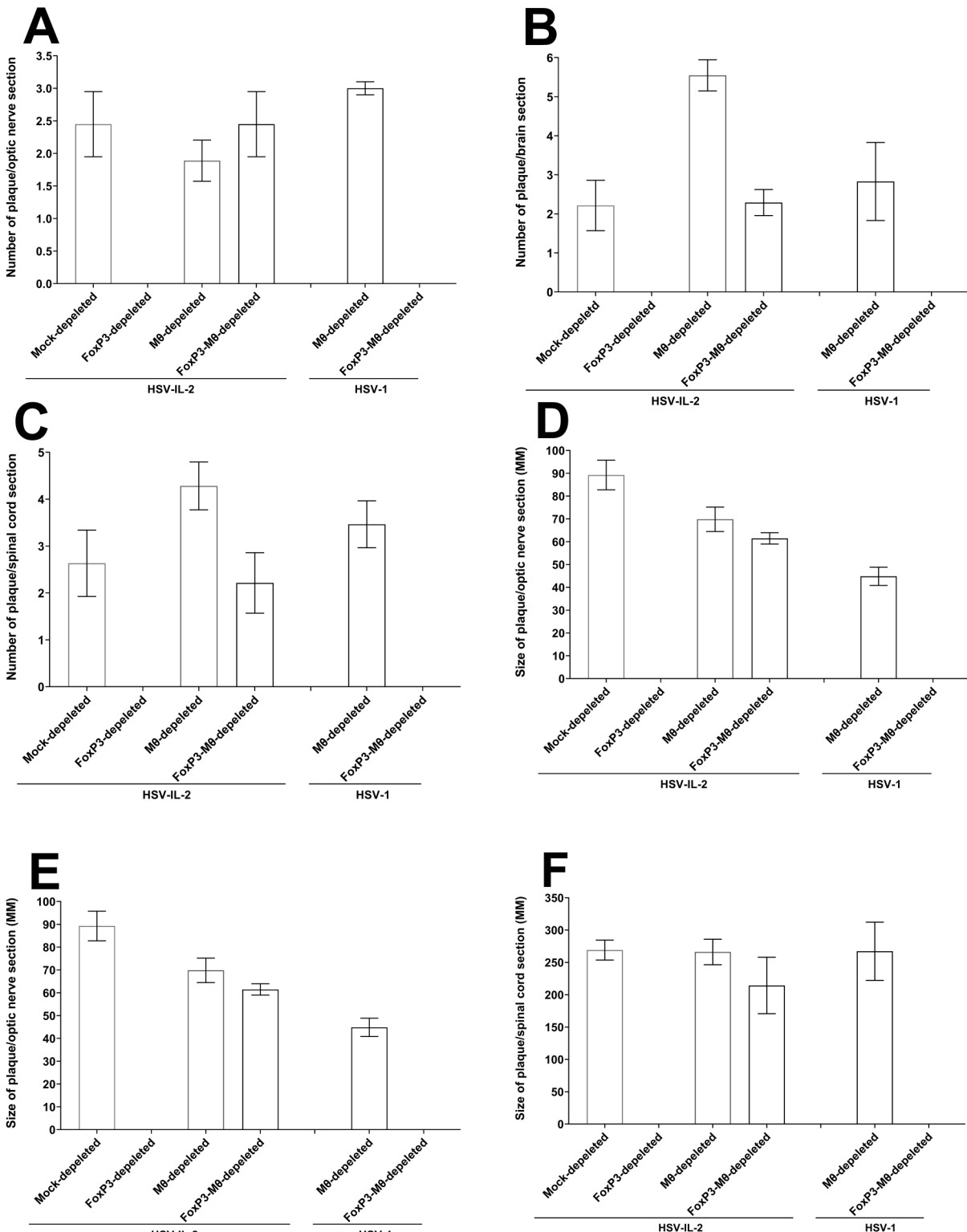

**Fig 4. Number and size of demyelination plaques in the CNS of infected mice.** The entire brain, spinal cord, and optic nerves of each of the five animals described in Fig 1 were sectioned, and every 5th slide of each tissue was stained. The number and size of demyelination plaques in the entire sections of the brain, spinal cord, and optic nerves were counted and measured, respectively. Panels **A-C** show the number of plaques per total sections

of optic nerve, spinal cord, and brain, while panels **D-F** represent the size of plaques per total sections of optic nerve, spinal cord, and brain. Data are presented as the mean number of demyelination and the size of demyelination plaques using a total of 150 sections for the brain and spinal cord and 30 sections for the optic nerve from 5 mice per group.

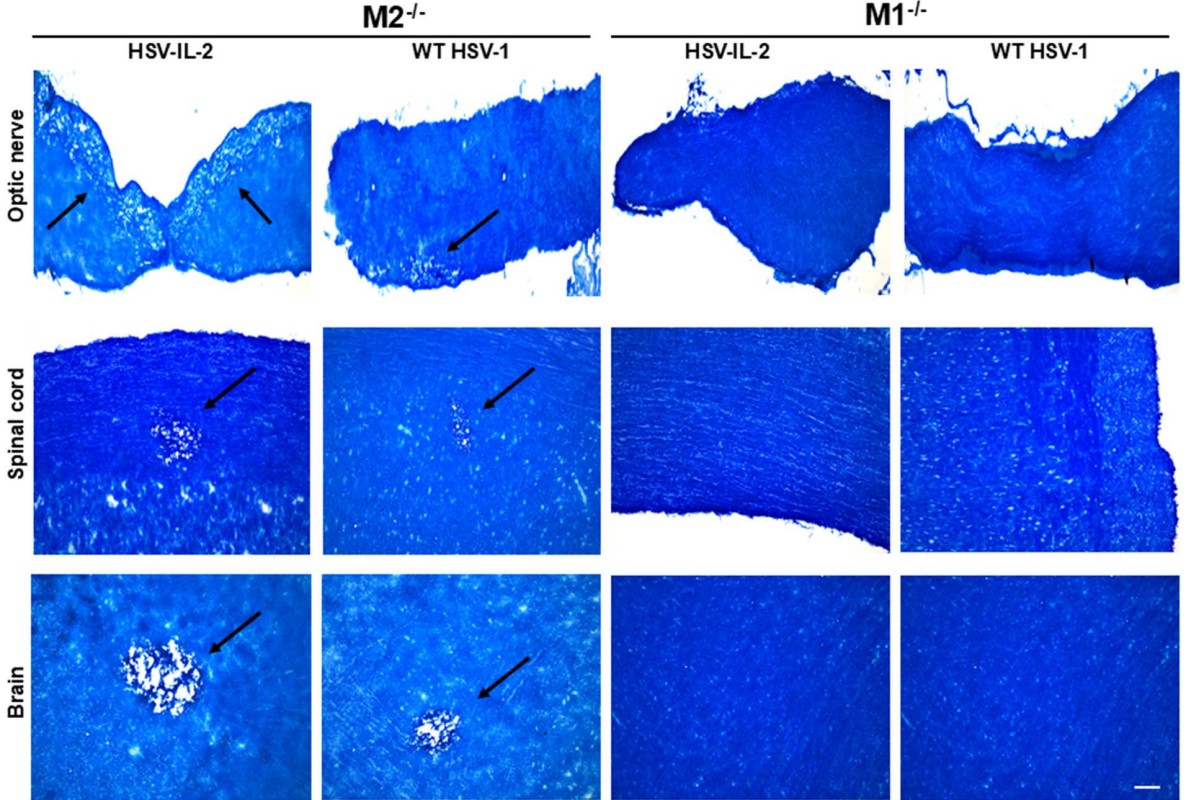

**Fig 5. Role of M1 and M2 macrophages in CNS demyelination.** M1[-/-] mice were infected with HSV-IL-2 or WT HSV-1 strain KOS, while M2[-/-] mice were infected ocularly with HSV-IL-2 or WT HSV-1 strain KOS as described in Materials and Methods. Representative photomicrographs of optic nerve, spinal cord, and brain sections on day 14 PI from both M1[-/-] and M2[-/-] infected mice are shown. The arrow indicates an area of demyelination (i.e., a plaque). Scale bar denotes 100 μm.

In contrast, the absence of M2 macrophages did not significantly alter HSV-1 infectivity from that of WT mice [53]. We also examined how the absence of M1 macrophages affects HSV-1 infectivity and have shown the importance of M1 macrophages specifically in controlling primary virus replication, eye disease, and survival in mice ocularly infected with WT HSV-1 [52]. Thus, identifying the roles of macrophages in our two alternative models of CNS demyelination requires rigorous and comprehensive analyses of macrophage responses to infection and their subsets. In this study, we demonstrated that depletion of macrophages using clodronate treatment ($Cl_2MDP$) caused CNS demyelination following infection with HSV-1. $Cl_2MDP$ has been extensively used to investigate the effects of the absence of macrophages on viral infection [4,59]. Thus, the impact of macrophage depletion on CNS pathology warrants careful consideration across different infection models.

Our published studies suggest that HSV plays a critical role in initiating myelin destruction in the presence of elevated levels of IL-2. IL-2 is a potent cytokine that promotes activation, proliferation, and persistence of various lymphocyte

subsets during differentiation, immune responses, and homeostasis [60]. In the HSV-IL-2 model, demyelination may be driven by either the innate or the adaptive arms of the immune responses. While macrophages are protective against CNS demyelination during WT HSV-1 infection [10], our previous work has shown that in HSV-1 IL-2 model, regardless of macrophage presence, IL-2 in HSV-IL-2 virus causes the suppression of IL-12p70, which leads to CNS demyelination by activating T-cell autoreactivity [18,61]. In WT HSV-1 infected mice, macrophages play a protective role against CNS damage, and co-depletion of macrophages and FoxP3 prevents CNS demyelination. In contrast, in HSV-IL-2-infected mice, FoxP3 depletion alone is sufficient to block demyelination, yet combined depletion of both macrophages and FoxP3$^+$Tregs paradoxically restores CNS demyelination. These contrasting outcomes highlight the pivotal role of IL-2 in shaping the immune environment and underscore that IL-2 alters the balance between immune regulation and autoreactivity in HSV-1-induced CNS pathology.

Next, we aimed to investigate the effects of IL-17 deficiency in our current model, building on our previous study where IL-17A$^{-/-}$ mice were resistant to HSV-IL-2-induced CNS pathology [32]. T$_H$17 cells, along with Tregs, have been implicated in both the pathogenesis of multiple autoimmune and inflammatory disorders as well as in tissue homeostasis via producing various proinflammatory and anti-inflammatory cytokines [62,63]. Similarly, in the experimental autoimmune encephalomyelitis (EAE) model of MS, T$_H$17 cells act as potent inducers of autoimmunity, with IL-17A$^{-/-}$ mice exhibiting significantly attenuated disease [64]. In the current study, we demonstrated that depletion of macrophages in the absence of IL-17A did not induce demyelination in HSV-IL-2-infected mice in contrast to depletion of macrophages in WT mice infected with WT-HSV-1. This finding aligns with our previous work, where IL-17A$^{-/-}$ mice were protected from HSV-IL-2-induced CNS demyelination, whereas mice lacking IL-17 receptors IL-17RC$^{-/-}$, IL-17RD$^{-/-}$, and IL-17RA$^{-/-}$RC$^{-/-}$ developed demyelination. Moreover, adoptive transfer of T cells from wild-type (WT) mice into IL-17A$^{-/-}$ mice or T cells from IL-17A$^{-/-}$ mice to Rag$^{-/-}$ mice induced CNS demyelination in infected mice [32]. These results indicate that IL-17A drives CNS demyelination independently of macrophages in WT mice.

Given the inherent plasticity and functional diversity of macrophages, which can dynamically adapt their phenotypes in response to environmental cues, our results indicate a substantial impact of IL-17A expression on macrophage plasticity in the presence of viral infection and IL-2 expression, leading to CNS demyelination. Macrophages are known to participate in T$_H$17 responses [65], and IL-17A influences cytokine secretion by macrophages [66]. Our results may suggest that IL-17A participates in M1 macrophage activation, and these novel interactions between IL-17 signaling and M1 inflammatory responses cause CNS demyelination. Previously, it was shown that M2 macrophages are essential for protection against parasitic infections and play roles in tissue repair and resolution of inflammation [67,68]. Similar to T$_H$17, Tregs may also switch M1/M2 macrophage type through multiple pathways affecting inflammation and tissue homeostasis [69].

Overall, in this study, we investigated the interconnected roles of T$_H$17, Treg, and macrophage subsets (M1 and M2) in CNS demyelination. Our findings reveal novel mechanistic insights into how immune signaling networks—particularly involving IL-17A and macrophage polarization—contribute to demyelinating pathology. Importantly, our data highlight a critical role for M1 macrophages in promoting CNS demyelination, driving T$_H$17 activation, and influencing FoxP3$^+$Treg dynamics. These insights may inform the development of targeted therapies aimed at modulating macrophage responses and IL-17 signaling to mitigate CNS damage in demyelinating diseases.

## Materials and methods

### Ethics statement

All animal procedures were performed in strict accordance with the Association for Research in Vision and Ophthalmology Statement for the Use of Animals in Ophthalmic and Vision Research and the NIH guide for Care and Use of Laboratory Animals (ISBN 0-309-05377-3). The Institutional Animal Care and Use Committee of Cedars-Sinai Medical Center (Protocols # 6134 and 9833) approved the animal research protocol.

## Viruses, cells, and mice

Plaque-purified HSV-1 strains, WT McKrae, dLAT2903 (parental virus for HSV-IL-2 virus and derived from McKrae virus), HSV-1 strain KOS, and HSV-1 recombinant virus expressing IL-2 (HSV-IL-2) were grown in rabbit skin (RS) cell monolayers in minimal essential medium (MEM) containing 5% fetal calf serum (FCS) as previously described [12,70]. McKrae and dLAT2903 viruses are virulent at an infectious dose of 2 X $10^5$ pfu/eye, whereas KOS and HSV-IL-2 viruses are attenuated strains. Female C57BL/6, FoxP3DTR, M1-/-, M2-/-, and IL-17A-/- mice aged 6–8 weeks were used. All mice were bred at Cedars-Sinai Medical Center. The absence of STAT1 expression in macrophages blocked M1 activation [71], while the absence of GATA3 in macrophages block M2 activation [53,72], thus throughout this study we are calling these mice M1-/- and M2-/-, respectively.

## Ocular infection

Mice were infected ocularly with 2 X $10^5$ pfu/eye of McKrae, dLAT2903, KOS, or HSV-IL-2 virus. Each virus was suspended in 2 μl of tissue culture media and administered as an eye drop [73]. Corneal scarification was not performed before infection with McKrae, dLAT2903, or HSV-IL-2 virus. For KOS virus infection, mice received 2 X $10^5$ pfu/eye of KOS virus with corneal scarification as we described previously [74]. Before corneal scarification and ocular infection, mice were anesthetized with ketamine + dexmedetomidine. Following anesthesia and ocular infection, buprenorphine was administered by subcutaneous injection. Buprenorphine was administered again in the morning after the infection.

## Macrophage depletion

Liposome-encapsulation of dichloromethylene diphosphonate ($Cl_2MDP$) was performed as we previously described [10,75]. To deplete macrophages, each mouse received 100 μl of the mixture intraperitoneally (IP) and subcutaneously. Macrophage depletion was conducted on days -5, -2, + 1, + 4, + 7, and +10 relative to ocular infection with HSV-1.

## Depletion of FoxP3

Female FoxP3DTR mice were depleted of their FoxP3 by treatment with diphtheria toxin (DT) as described previously [10,76]. Briefly, the mice were administered DT at 72 and 24 h before ocular infection, followed by 5 additional treatments on days +1, + 3, + 5, + 7, and +9 post infection.

## Isolation of spleen cells for flow cytometry

Spleens from HSV-IL-2 or McKrae (control) infected mice were harvested on day 3 post infection (PI). Single-cell suspensions from individual mouse spleen were prepared as described previously [12]. Monoclonal antibodies used for flow cytometry were purchased from BD Biosciences (San Diego, CA) and BioScience (San Diego, CA). Cell surface and intracellular staining of single cell suspensions of splenocytes from an individual mouse was accomplished according to the manufacturer's instructions. Antibodies included anti-CD4-FITC (clone L3T4), anti-CD8a-PE (clone 53-6.7), and anti-FoxP3-Pacific blue (clone FJK-16s). Three-color flow cytometric analyses of splenocytes were performed using a FACScan instrument (BD Biosciences, San Jose, CA). The percentage of cells stained with mAbs was calculated by using FlowJo software with forward/side scatter gating of MNCs or splenocyte preparations. Controls included non-relevant isotype-matched antibody, no primary antibody, or no secondary antibody alone. A minimum of 1 X $10^4$ events was acquired based on a live cell gate.

## Tissue preparation

Optic nerves (ONs), brains, and spinal cords (SCs) of experimental and control mice were removed on day 14 PI, embedded in optimal cutting temperature compound (OCT, Tissue-Tek, Sakura) for cryo-sectioning, and stored at -80˚C as we described previously [12].

## Demyelination morphometry

Transverse sections of brains, spinal cords, and optic nerves, 10 µm thick (spaced 50µm apart), were sectioned using a Leica CM3050S cryostat, air-dried overnight, and fixed in acetone for 3 min at 25℃ [77]. The presence or absence of demyelination in the CNS of five infected mice per group was evaluated using luxol fast blue (LFB) staining as we described previously [12]. Every tenth section of the brain and spinal cord was stained for LFB. The number and size of visible, pronounced demyelinated areas per section were counted. Demyelination in each section was confirmed by monitoring adjacent sections. The number, size, and shape of plaques on multiple fields were evaluated by investigators who were blinded to the treatment groups using serial sections of CNS tissues. The amount of myelin loss in the stained sections of brains, spinal cords, and optic nerves was measured using the NIH Image J software analysis system. The areas of demyelination (clear-white) to normal tissue (blue) were quantified using 150 random sections from the brain and spinal cords or 30 sections from the optic nerves of each animal. Demyelination in each section was confirmed by monitoring adjacent sections. The percentage of myelin loss was calculated by dividing the lesion size by the total area for each section. A total of 5 mice per group were evaluated in our experiments for three different tissues, as stated in the results section, and the groups showing demyelination were all 100% demyelination positive.

## Statistical analyses

For all statistical tests, p-values less than or equal to 0.05 were considered statistically significant and marked by a single asterisk (*). P-values less than or equal to 0.001 were marked by double asterisks (**). To analyze the presence or absence of demyelination in experimental and control groups, Fisher's exact tests were performed using Instat (Graph-Pad, San Diego). An examiner blinded to sample identities performed all analyses, and code was not broken until all analyses were completed. For all analyses, we set alpha levels at 0.05.

## Supporting information

**S1 Data.**
(PDF)

## Author contributions

**Conceptualization:** ujjaldeep Jaggi, Homayon Ghiasi.

**Data curation:** ujjaldeep Jaggi, Satoshi Hirose, Shaohui Wang.

**Formal analysis:** Satoshi Hirose, Shaohui Wang, Homayon Ghiasi.

**Funding acquisition:** Homayon Ghiasi.

**Investigation:** Homayon Ghiasi.

**Methodology:** ujjaldeep Jaggi, Satoshi Hirose, Shaohui Wang.

**Project administration:** Homayon Ghiasi.

**Resources:** Homayon Ghiasi.

**Supervision:** Homayon Ghiasi.

**Validation:** ujjaldeep Jaggi, Satoshi Hirose, Homayon Ghiasi.

**Visualization:** Satoshi Hirose, Homayon Ghiasi.

**Writing – original draft:** ujjaldeep Jaggi, Homayon Ghiasi.

**Writing – review & editing:** ujjaldeep Jaggi, Homayon Ghiasi.

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
