## [Decision Letter · Decision Letter 0]

14 Oct 2025

PPATHOGENS-D-25-02100

Synergistic actions of M1 macrophages and TH17 in CNS demyelination

PLOS Pathogens

Dear Dr. Ghiasi,

Thank you for submitting your manuscript to PLOS Pathogens. After careful consideration, we feel that it has merit but does not fully meet PLOS Pathogens's publication criteria as it currently stands. Therefore, we invite you to submit a revised version of the manuscript that addresses the points raised during the review process.

Please submit your revised manuscript within 30 days. If you will need more time than this to complete your revisions, please reply to this message or contact the journal office at plospathogens@plos.org. Please include the following items when submitting your revised manuscript:

We look forward to receiving your revised manuscript.

Kind regards,

Deepak Shukla

Academic Editor

PLOS Pathogens

Donna Neumann

Section Editor

PLOS Pathogens

Sumita Bhaduri-McIntosh

Editor-in-Chief

PLOS Pathogens

orcid.org/0000-0003-2946-9497

Michael Malim

Editor-in-Chief

PLOS Pathogens

orcid.org/0000-0002-7699-2064

**Journal Requirements:**

1) We note that your Data Availability Statement is currently as follows: "All relevant data are within the manuscript.". Please confirm at this time whether or not your submission contains all raw data required to replicate the results of your study. Authors must share the “minimal data set” for their submission. PLOS defines the minimal data set to consist of the data required to replicate all study findings reported in the article, as well as related metadata and methods (https://journals.plos.org/plosone/s/data-availability#loc-minimal-data-set-definition).

**Reviewers' Comments:**

Reviewer's Responses to Questions

**Part I - Summary**

Reviewer #1: In this well-designed and documented study, the authors investigated the roles of TH17, Treg, and macrophage subsets (M1 and M2) in murine models of CNS demyelination following HSV-1 ocular infection. The study provides novel observations concerning the contributions of immune signaling networks, particularly those involving IL-17A and macrophage polarization, to demyelinating pathology. The study highlights a critical role for M1 macrophages in promoting CNS demyelination, driving TH17 activation, and influencing FoxP3⁺ Treg dynamics. These findings are novel and highly relevant to the field.

The study is generally well documented, however, in my opinion, the clarity of the presentation needs improvement in a few areas.

Reviewer #2: The manuscript titled “Synergistic actions of M1 macrophages and TH17 in CNS demyelination” by Jaggi and co-workers explores the interplay between macrophages, FoxP3+ regulatory T cells, and IL-17 in CNS demyelination following ocular infection with HSV-1. The authors demonstrate that infection with an IL-2-expressing HSV-1 strain leads to CNS demyelination in mice depleted of both macrophages and FoxP3+ cells, whereas WT HSV-1 infection does not induce demyelination under the same conditions. Interestingly, macrophage depletion alone during WT HSV-1 infection was sufficient to trigger demyelination, suggesting a protective role for macrophages in this context. To further investigate the contribution of macrophage subsets, the authors utilized M1- and M2-deficient mice. They report that the absence of M1 macrophages prevented demyelination, while M2-deficient mice exhibited CNS demyelination in the HSV-IL-2 strain, supporting a pathogenic role for M1 and a protective role for M2 macrophages. Additionally, depletion of macrophages in IL-17-deficient mice infected with HSV-IL-2, did not lead to demyelination, indicating that IL-17 contributes to demyelination in a macrophage-dependent manner and the absence of IL-17-macrophage is beneficial.

Overall, the findings are novel and provide valuable insights into the distinct and non-redundant roles of macrophages, Th17, and FoxP3+ Tregs in modulating CNS pathology during HSV-1 infection. The study is well-conceived and contributes meaningfully to our understanding of neuroinflammatory mechanisms.

Reviewer #3: Summary: Multiple sclerosis is a major demyelinating neurologic disease that is associated with an immunopathologic origin. The Ghiasi laboratory has developed an intriguing mouse model for demyelinating neurologic diseases in which C57BL/6 mice are inoculated corneally with an IL-2-expressing recombinant HSV-1 (HSV-IL-2). These animals subsequently show demyelination in optic nerves, brain, and spinal cord. Because previous work has shown that depletion of macrophages induces demyelination in HSV-IL-2-infected mice, a series of studies were performed to assess further a role for macrophages in demyelination with emphasis on FoxP3 (T regulatory cells or Treg cells) and IL-17A participation. Results revealed (i) combined depletion of macrophages and FoxP3 triggered CNS demyelination in HSV-IL-2-infected mice but prevented demyelination in WT HSV-1-infected mice; (ii) depletion of macrophages in IL-17A-deficient mice did not restore CNS demyelination in HSV-IL-2-infected mice; and (iii) M1 macrophages are key drivers of plaque formation. The authors conclude that FoxP3, IL-17A, and macrophage subsets play distinct and non-redundant roles in modulating CNS demyelination during HSV-1 infection and suggest that targeting M1 macrophage activation might be a new therapeutic approach for limiting CNS demyelination.

Review: This investigation summarizes the findings of a series of relatively straightforward studies that add new and important knowledge regarding the relative roles of macrophages, FoxP3, and IL-17A during CNS demyelination using a well-established HSV-1 mouse model of the CNS demyelination. The experiments appear to be performed carefully with rigor and data are convincing. Although the manuscript is already strong in form and content, some attention to the following observations might further strengthen the manuscript.

1. The Title is far too brief and uninformative and needs to be revised to inform potential readers that the investigation and conclusions as stated involve a mouse model of HSV-1 CNS demyelination.

2. Are paragraphs 2 and 3 of the Introduction section necessary? They summarize previous work on IL-12p35 and IL-12p40 as well as innate lymphoid cells that distract from the true focus of the studies summarized in the manuscript.

3. Although the inclusion of histopathologic figures in the Results section are convincing, the authors should provide quantification perhaps as percent animals showing demyelination versus total number of animals evaluated. Simply put, of the animals used per experimental group, how many were positive and negative for CNS demyelination? If 100%, then communicate this to the reader. Either provide this information in the Results section text or provide a table summarizing the results quantitatively.

4. Figure 4 is very blurry and difficult to read.

5. Is data on size and number of plaques (Fig 4) derived from the same experiment shown in Fig 1, or are these separate experiments?

6. The authors should caution against the statement that the KOS strain of HSV-1 is “avirulent” which means an absolute lack of disease/pathology. When compared with the McKrae strain, the KOS strain is weakly virulent.

**Part II – Major Issues: Key Experiments Required for Acceptance**

Reviewer #1: No major issues identified

Reviewer #2: Not applicable

Reviewer #3: None

**Part III – Minor Issues: Editorial and Data Presentation Modifications**

Reviewer #1: In the Demyelination morphometry section of the Materials and Methods section, the authors state that “The presence or absence of demyelination in the CNS of five infected mice per group was evaluated using luxol fast blue (LFB) staining as we described previously [14].”

It would seem important to provide here some additional information based on previous published work that LFB staining in the used mouse models at the studied time (14 days PI with virus) and in the studied CNS tissues LFB staining alone is sufficient to identify demyelinating lesions. Brief description of these lesions in the used models as far as myelin loss, axonal preservation and presence or absence of virus/viral products would be important for the better understanding of the model.

Due to repetition of text, correction is needed to the sentence in lines 83—86, “Although, IL-12p70 is produced by a variety of immunocompetent cells, including monocytes, dendritic cells, neutrophils, and B cells [11, 12], in addition to macrophages IL-12p70 is produced by a variety of immunocompetent cells, including monocytes, dendritic cells, neutrophils, and B cells [11, 12],….

Reviewer #2: I have the following specific comments for the authors:

Line 83-86, delete the duplicated sentence “IL-12p70 is produced by a……….B cells”

Fig 2A-B bar graphs, the X axis is labeled as HSV-IL-2, while the other bar is labeled as control. For clarity and consistency, I suggest relabeling control as WT HSV-1. Also, if available, it would strengthen the data presentation if authors include representative flow cytometry dot plots for the analyzed cell populations.

Line 179-182, the sentence seems incomplete; the authors should correct/reframe this sentence by citing ….…HSV-IL-2 infected mice (for 3 A-C) and …….WT HSV-1 infected mice for Fig 3D-F.

Lines 182-183, the authors make a statement about a comparison between IL17A KO mice infected with HSV-IL-2 and WT HSV-1 strains. However, it appears that this data is missing. Fig. 3 only shows a comparison of IL-17 KO-macrophage-depleted mice with HSV-IL-2 infection with WT-HSV1-infected WT mice with macrophage depletion. Authors should verify if this is an error and show the data or correct the statement.

In Fig. 3, some of the panel labels (A-F) are on micrographs, while others are outside, depending on the size of the tissue section. It would be better if the authors keep these labels out of micrographs.

Figure 4 (A-F) axis labels are tiny and unreadable. Can authors improve the visualization for better clarity?

For Figs. 4A-F, the WT virus is simply labeled as HSV-1, whereas in all other figures, it is labeled as WT HSV-1. The authors may want to consider changing this to maintain consistency with the rest of the figures.

For Fig. 4, the statistics are described in the results text but are not included in the figures. It would be easier for readers if they show statistics (either * or p values, as they did in Fig. 2), on figures/bars as well.

Lines 623-624, the figure legend statement is incomplete, and the mentions for the viral strains (HSV-1-IL-2 vs. WT HSV-1) are missing after the statements ……..ON, SC, and brain….from ……

In Fig. 5, it appears that scale (black) bars are present on some micrographs but not all. For consistency and clarity, please ensure that scale bars are included on all images. Additionally, consider removing the magnification (X) from the figure legends and adding a scale bar to all micrograph panels, as scale bars provide a more precise and interpretable reference.

Reviewer #3: Figure 4 is very blurry and difficult to read.

PLOS authors have the option to publish the peer review history of their article (what does this mean? ). If published, this will include your full peer review and any attached files.

**Do you want your identity to be public for this peer review?** For information about this choice, including consent withdrawal, please see our Privacy Policy .

Reviewer #1: No

Reviewer #2: No

Reviewer #3: No

**Figure resubmission:**
---

## [Decision Letter · Decision Letter 1]

5 Nov 2025

Dear Dr. Ghiasi,

We are pleased to inform you that your manuscript 'Divergent roles of macrophage subsets, FoxP3, and IL-17A in HSV–1–induced CNS pathology' has been provisionally accepted for publication in PLOS Pathogens.

Best regards,

Deepak Shukla

Academic Editor

PLOS Pathogens

Donna Neumann

Section Editor

PLOS Pathogens

Sumita Bhaduri-McIntosh

Editor-in-Chief

PLOS Pathogens

orcid.org/0000-0003-2946-9497

Michael Malim

Editor-in-Chief

PLOS Pathogens

orcid.org/0000-0002-7699-2064

Reviewer Comments (if any, and for reference):

Reviewer's Responses to Questions

**Part I - Summary**

Reviewer #1: The revised manuscript contains all the suggested changes and, in my opinion, should be accepted for publication.

Reviewer #2: The authors have satisfactorily addressed the concerns raised during the review process and have revised the figures and data as requested.

Reviewer #3: The authors have responded thoughtfully and completely to all previous concerns and questions. The revised manuscript is now far stronger than the original.

**Part II – Major Issues: Key Experiments Required for Acceptance**

Reviewer #1: None identified

Reviewer #2: NA

Reviewer #3: None

**Part III – Minor Issues: Editorial and Data Presentation Modifications**

Reviewer #1: None identified

Reviewer #2: NA

Reviewer #3: None

PLOS authors have the option to publish the peer review history of their article (what does this mean? ). If published, this will include your full peer review and any attached files.

**Do you want your identity to be public for this peer review?** For information about this choice, including consent withdrawal, please see our Privacy Policy .

Reviewer #1: No

Reviewer #2: No

Reviewer #3: No

---

## [Editor Report · Acceptance letter]

Dear Dr. Ghiasi,

We are delighted to inform you that your manuscript, "Divergent roles of macrophage subsets, FoxP3, and IL-17A in HSV–1–induced CNS pathology," has been formally accepted for publication in PLOS Pathogens.

Best regards,

Sumita Bhaduri-McIntosh

Editor-in-Chief

PLOS Pathogens

orcid.org/0000-0003-2946-9497

Michael Malim

Editor-in-Chief

PLOS Pathogens

orcid.org/0000-0002-7699-2064